# Connexin-46 Contained in Extracellular Vesicles Enhance Malignancy Features in Breast Cancer Cells

**DOI:** 10.3390/biom10050676

**Published:** 2020-04-28

**Authors:** Rodrigo A. Acuña, Manuel Varas-Godoy, Viviana M. Berthoud, Ivan E. Alfaro, Mauricio A. Retamal

**Affiliations:** 1Universidad del Desarrollo, Centro de Fisiología Celular e Integrativa, Facultad de Medicina Clínica Alemana, Universidad del Desarrollo, Santiago 7780272, Chile; 2Universidad del Desarrollo, Programa de Comunicación Celular en Cáncer, Instituto de Ciencias e Innovación en Medicina (ICIM), Facultad de Medicina Clínica Alemana, Santiago 7780272, Chile; 3Universidad del Desarrollo, Instituto de Ciencias e Innovación en Medicina (ICIM), Facultad de Medicina Clínica Alemana Universidad del Desarrollo, Santiago 7780272, Chile; ialfaro@udd.cl; 4Cancer Cell Biology Lab., Centro de Biología Celular y Biomedicina (CEBICEM), Facultad de Medicina y Ciencia, Universidad San Sebastián, Santiago 7780272, Chile; manuel.varas@uss.cl; 5Department of Pediatrics, University of Chicago, Chicago, IL 60637, USA; vberthou@peds.bsd.uchicago.edu; 6Fundación Ciencia & Vida, Avenida Zañartu #1482, Ñuñoa, Santiago 7780272, Chile

**Keywords:** gap junction channels, connexin 46, extracellular vesicles, breast cancer

## Abstract

Under normal conditions, almost all cell types communicate with their neighboring cells through gap junction channels (GJC), facilitating cellular and tissue homeostasis. A GJC is formed by the interaction of two hemichannels; each one of these hemichannels in turn is formed by six subunits of transmembrane proteins called connexins (Cx). For many years, it was believed that the loss of GJC-mediated intercellular communication was a hallmark in cancer development. However, nowadays this paradigm is changing. The connexin 46 (Cx46), which is almost exclusively expressed in the eye lens, is upregulated in human breast cancer, and is correlated with tumor growth in a Xenograft mouse model. On the other hand, extracellular vesicles (EVs) have an important role in long-distance communication under physiological conditions. In the last decade, EVs also have been recognized as key players in cancer aggressiveness. The aim of this work was to explore the involvement of Cx46 in EV-mediated intercellular communication. Here, we demonstrated for the first time, that Cx46 is contained in EVs released from breast cancer cells overexpressing Cx46 (EVs-Cx46). This EV-Cx46 facilitates the interaction between EVs and the recipient cell resulting in an increase in their migration and invasion properties. Our results suggest that EV-Cx46 could be a marker of cancer malignancy and open the possibility to consider Cx46 as a new therapeutic target in cancer treatment.

## 1. Introduction

In higher organisms, intercellular communication is a complex process, vital to maintaining normal tissue functions. Under physiological conditions, gap junction channels (GJC) are one of the most important mechanisms involved in cell-to-cell communication. A GJC is formed by the head-to-head docking of two hemichannels, which in turn are formed by six subunits of transmembrane proteins called connexins (Cx) [1]. Channels formed by Cx allow the diffusional movement of ions, metabolites, and signaling molecules with a permeability up to 1.2 kDa [2]. Twenty one different Cx genes have been identified in humans [3], with expression in almost all cell types and tissues [4,5,6]. The importance of Cxs in human diseases, has gained great attention because hemichannel and GJC malfunctioning correlates with the progression of several diseases including cancer, deafness, skin disorders, atrial fibrillation, and cataracts [7,8].

Under normal conditions, cells are communicating with their neighboring partners in order to maintain the proper tissue function. However, in cancer, the cellular communication is altered [5,9]. In particular, the loss of intercellular GJC communication by a reduction of Cx expression has been considered as a hallmark of cancer development for many years [10,11,12,13,14,15,16]. However, the role of Cxs in cancer has been shown to be very complex and still remains an open question [17]. For example, Cx43, which under normal conditions is present widely present in the human body, has been related to the progression of multiple types of cancer [18,19,20], not only in its role in local communication but also in a long distance communication mediated by extracellular vesicles (EVs) [20]. These EVs are lipid bilayer membranes with an endocityc origin and with a size range ranging between 40–150 nm [21,22,23]. These EVs are released into the extracellular milieu by normal and tumor cells, and act as natural carriers of “biological information”, which include molecules such as DNA, mRNA, microRNAs (miRNAs), proteins, and lipids [23,24,25].

It is well known that “in transit” EVs can be found in different human fluids, including blood, plasma, saliva, urine, synovial fluid, and amniotic fluid [26]. These EVs eventually will find by an unknown mechanism a recipient cell where it will deliver its content [27,28]. EVs can be incorporated into the cells by different mechanisms [29,30]. However, independent of these mechanisms, the final results are the release of their cargos into the recipient cell, where they will modify its behavior [31,32,33,34]. For example, in in vitro experiments cancer-associated fibroblasts show the transference of specific miRNAs through EVs to breast cancer cells, increasing their capacity to form mammospheres, cancer stem cells or to undergo ephitelial-mesenchymal transition (EMT) processes [35]. However, not only miRNAs can be transferred, other studies have also shown protein transference [36]. For example, αVβ3 integrin is up-regulated in cancer and promotes migratory and metastatic phenotypes [37], EVs released from prostate cancer transfer αVβ3 to non-tumorigenic cells and increase migration in the recipient cell [36]. As a result of these evidences, and many others, currently EVs are strongly associated with the malignancy of cancer cells, and are considered key players in cancer development and aggressiveness [25].

Unlike Cx43, Cx46 has been much less studied, probably because under normal conditions, it is mostly found in the mammalian lens [38]. However, low (or very low) levels of this protein also has been found in rat bones [39], rat lungs [40], mouse hearts [41], and mouse astrocytes [42]. Cx46 is implicated in lens homeostasis, and deregulation of its hemichannels and GJCs strongly correlates with cataract formation [43,44,45]. In addition to its traditional role in the lens, Cx46 also have been implicated in cancer progression [11,12,46,47]. For example, there is an increase in Cx46 protein expression in human breast cancer [12], and have been correlated with the establishment of cancer stem cells in glioblastoma [11,46]. In agreement with the role of Cx46 in cancer progression, siRNA knockdown of Cx46 drastically decreases the size of retinoblastoma-induced tumors in a xenograft mouse model [48].

Despite growing evidence of the importance of Cx46 and EVs in cancer development and aggressiveness, there is no information about the presence of Cx46 in EVs derived from breast cancer cells, and its possible implication in long-distance communication. Therefore, in this work we studied whether Cx46 is present in EVs released from a cell line derived from human breast cancer, and whether or not the presence of Cx46 in the EVs modulate their effects upon recipient cells. To this end, we created a stable MCF-7 human breast cancer cell line overexpressing human Cx46 attached to green fluorescent protein (GFP) (MCF-7Cx46-GFP). We observed that, the Cx46 overexpression in MCF-7 cells resulted in the incorporation of Cx46 in MCF-7-derived EVs (EVs-Cx46). The expression of Cx46 in MCF-7 cells, resulted in an increase of the number of EVs released compared with low expression MCF-7 cells, and also in an increase in the EV-recipient cell interaction and the transfer of “information” between the EVs and breast cancer MCF-7 cells. 

## 2. Materials and Methods

### 2.1. Cells

Human breast cancer cell line, MCF-7 (HTB22, ATCC) and human cervical cancer cell line, HeLa, were obtained from the American Type Culture Collection (ATCC). MCF-7-Cx46GFP and HeLa-Cx46GFP were obtained by stable transfection of human connexin46-plasmid pcmV6-AC-GFP (Origene, Rockville, MD, USA), and selected by Neomycin (Sigma-Aldrich Saint Louis, MO, USA) resistance given by the plasmid. All cell lines were maintained in Dulbecco’s modified Eagle’s medium (Life Technologies, Carlsbad, CA, USA), supplemented with 10% of heat-inactivated fetal bovine serum (Life Technologies, Carlsbad, CA, USA) and penicillin/streptomycin (100 U/mL Penicillin, 100 µg/mL Streptomycin) (Gibco). Cells were maintained at 37 °C in a 5% CO_2_ humidified incubator.

### 2.2. Extracellular Vesicles Isolation

EVs were obtained by ultracentrifugation. Briefly, MCF-7, MCF-7-Cx46GFP, HeLa, and HeLa-Cx46GFP cells were cultured in Dulbecco´s modified Eagle´s medium (DMEM) (Life Technologies, Carlsbad, CA, USA) supplemented with 10% fetal bovine serum. After achieved a confluence of 70%, the medium was replaced by DMEM supplemented with 10% of exosome depleted fetal bovine serum. Forty eight hours later, the supernatants were collected and subjected to different centrifugation steps at 4 °C, starting with centrifugation at 300× *g* for 5 min, followed by 1500× *g* for 10 min, 10,000× *g* for 30 min, and finally supernatants were ultracentrifuged at 100,000× *g* for 90 min. The pellet obtained was resuspended in PBS for further analysis.

### 2.3. Western Blot

Briefly, cells and EVs were lysed in RIPA buffer supplemented with protease inhibitors (Roche). The protein concentration was determined using a protein assay kit (ThermoFisher Scientific, Waltham, MA, USA) and read in a Qubit 3.0 Fluorometer (ThermoFisher Scientific, Waltham, MA, USA). To characterize EVs markers, 30 μg of total protein from EVs and from the respective cells were resolved on 12% SDS PAGE gel by PAGE and transferred to Nitrocellulose membrane (Bio-Rad, Hercules, CA, USA). Membranes were incubated with one of the following primary antibodies: Alix, flotillin, CD9, Calnexin, GM130 (All from Cell Signaling; 1:2000) or Cx46 (Santa Cruz Biotechnology; 1:500). All secondary antibodies were horse-radish protein (HRP) conjugated (Abcam). Protein bands were detected using Immobilon Forte western HRP substrate (Millipore, Burlington, MA, USA) and visualized with LI-COR C-Digit Chemiluminescense Western Blot Scanner systems (LI-COR, Inc, Lincoln, USA).

### 2.4. Nanoparticle Tracking Analysis (NTA)

EVs isolated from MCF-7, MCF-7-Cx46-GFP, and HeLa cells were subjected to Nanoparticle tracking analysis (NTA), using a NanoSight NS300 instrument (Malvern Instruments Ltd., Amesbury, UK). Settings were optimized and kept constant between samples. Each video was analyzed and the mean, mode, median, and estimated concentration for each particle were calculated. Data were processed using NTA 2.2 analytical software (Malvern Instruments Ltd., Amesbury, UK).

### 2.5. Transmission Electron Microscopy (TEM)

EVs were deposited on Formvar-carbon coated grids (TED Pella, Mountains Lake, CA, USA), after 1 min of adsorption the excess was removed with absorbent paper and contrasted with uranyl acetate pH 7.0 for 1 min, excess was removed and dried for 5 min at 60 °C. The specimens were observed using a Philips Tecnai 12 BioTWIN electron microscope (FEI) at 80 kV (Unidad de microscopia avanzada UC (UMA), Pontificia Universidad Católica de Chile, Santiago, Chile). Immunogold labeling was performed as follows: EVs were adsorbed as described above on Formvar-carbon coated grids, and permeabilized with 0.1% saponin, rinsed in PBS, and blocked with 0.5% of BSA. Grids were then incubated with mouse anti-Cx46 antibodies (Santa Cruz Biotechnology, USA), rinsed with PBS, and labeled with a secondary antibody conjugated to 10 nm gold particles (Abcam, Cambridge, UK). Specimens were contrasted and observed as described above using a Philips Tecnai 12 BioTWIN electron microscope (FEI) at 80 kV.

### 2.6. Membrane Labeling of EVs

Ultracentrifugation-purified EVs were incubated with PKH26 (Sigma-Aldrich, Saint Louis, MO, USA), PKH26 was diluted in 100 μL of diluent C to a final concentration of 8 μM (dye solution). Ten μg of EVs were diluted with 80 μL of diluent C, added to the dye solution, and incubated for 5 min with mixing by gentle pipetting. Excess dye was bound with 100 μL of 10% EVs-depleted fetal bovine serum. Then EVs were diluted with PBS and subjected to ultracentrifugation for 2 h at 100,000×*g*. The pellet was resuspended in PBS. A solution without EVs was labeled in parallel with the EV preparation and used as a negative control. 

### 2.7. Microscopy Imaging of EV Uptake

Confocal analysis was performed to visualize if EVs were incorporated by MCF-7 and MCF7-Cx46 recipient cells, confocal analysis was performed. Image stacks were obtained from MCF-7 and MCF-7Cx46-GFP cells incubated with EVs derived from high expression MCF-7Cx46-GFP (EVs-Cx46) and EVs derived from low expression MCF-7 cells (EVs-sans-Cx46). MCF-7 cells were incubated for 3 h at 37 °C or 4 °C with EVs-Cx46 and EVs-sans-Cx46. After that, cells were washed with 4 °C PBS to eliminate non-incorporated EVs, fixed, and permeabilized with Triton X-100 (0.2% v/v), incubated with CellMask Deep Red (which strains the entire cell) (Thermofisher, Waltham, MA, USA) at 1× final concentration for 30 min, washed with PBS and mounted using Dapi-Fluoromount (Electron Microscopy Sciences, Hatfield, USA), and visualized in an FV1200 Olympus confocal microscopy. Image were acquired in a 60x AN1.4 oil immersion objective at 2048 × 2048 pixel resolution with a Z step of 10 × 0.5 μm using the corresponding excitation and emission settings for; PKH26 (EVs) (551ex, 567em), Cell Mask Deep Red (stains the entire cell) (Thermofisher, Waltham, MA, USA) (649ex, 666em), and GFP (Cx46-GFP) (450ex, 480em). Z-stack projections were obtained and intensity correlation analysis was carried out using the Fiji software (NHI, Bethesda, MD, USA) [49,50].

### 2.8. Extracellular Vesicles DNA Loading

EVs isolated from MCF-7 and MCF-7Cx46-GFP cells respectively, were loaded with a plasmid encoding green fluorescent protein (GFP) according to the protocol published by Soares. A et al. [20] with a few modifications. Briefly, 5 μg of plasmid were mixed with 10 μg of Lipofectamine 2000 (Life Technologies) in Opti-MEM for 10 min; then 25 μg of EVs were incorporated to the mix and incubated for 30 min at room temperature. To eliminate the non-incorporated plasmid, 5 U of DNAse I (New England Biolabs) were added and incubated for 30 min at 37 °C. EVs were filtered through a 100-kDa filter (Millipore, Burlington, USA). Then, 5 × 10^5^ of MCF-7 recipient cells were seeded on 35 mm dishes, 24 h later 2.5 μg of DNA-loaded EVs-sansCx46 or EVs-Cx46 were co-cultured with recipient cells for 45 min. After that, the medium was replaced with EVs-depleted medium and left for 24 h to analyze protein expression. GFP expression was evaluated by flow cytometry and fluorescence microscopy. For flow cytometry, MCF-7 cells were detached with trypsin, rinsed with PBS three times and resuspended in PBS. Cytometry was performed using a FACSCantoII (BD Biosiences, Franklin Lakes, USA). For fluorescent microscopy, cells were fixed with 4% PFA, mounted with Fluoromount (Sigma-Aldrich, Saint Louis, MO, USA) and visualized using a Nikon Eclipse Ti-U inverted microscope (Nikon, Melville, NY, USA).

### 2.9. Measurement of EVs Uptake

EVs labeled with PKH26 were incubated with EVs derived from MCF-7 and EVs-Cx46 isolated from MCF-7Cx46-GFP cells for 3 h. Then, the cells were rinsed with PBS (to eliminate unbound EVs), fixed and stained with DAPI. Cells were visualized on an Olympus Fluoview FV10i confocal microscope (Centro de medicina regenerativa (CMR), Universidad del Desarrollo, Santiago, Chile). To evaluate EV uptake, the cells were incubated at 37 °C (as a control condition), at 4 °C (to inhibit EVs endocytosis), or in the presence of 200 μM of lanthanum chloride at 37 °C (to block connexin hemichannel activity).

### 2.10. Migration Assay

Briefly, 6.0 × 10^5^ cells were plated in 35-mm culture dishes. After 24 h, the cell monolayers were wounded with a 200-μL sterile pipette tip; cellular debris was washed with PBS and the medium was replaced with Dulbecco´s modified Eagle´s medium (Life Technologies, Waltham, MA, USA), supplemented with 10% FBS. The wound was photographed at different times (0, 2, 4, 8, and 16 h) in a Nikon Eclipse Ti-U inverted microscope (Centro de Fisiología Celular e integrativa, Universidad del Desarrollo, Santiago, Chile). The area of wound closure area was measured using the NIS-Element AR 4.3 Software (Nikon, Melville, NY, USA).

### 2.11. Transwell Invasion Assay

Transwell chambers with 24 wells and 8 μm polycarbonate membrane (Corning, New York, USA) were used following the manufacturer’s protocol. Upper chambers were coated with 100 µL of DMEM-diluted Geltrex matrix (Gibco) and incubated at 37 °C for 6 h to allow the gel to solidify. Cultured cells were detached using 0.25% trypsin-EDTA solution. Cells were counted in a Neubauer chamber. Then, 1 × 10^5^ cells were seeded into the upper chambers in 200 µL of serum-free media. A total of 500 µL of complete medium were added to the lower chamber as a chemo attractant. After 12 h, cells remaining in the upper side of the polycarbonate membrane were removed with cotton swabs. Bottom chambers containing invasive cells were washed (twice with PBS), fixed in 100% methanol,and stained with DAPI (5 μm) for 5 min. Ten visual fields of each insert were randomly chosen, and photographed at 40Xmagnification. The number of cells/field was quantified using ImageJ software (NIH, Bethesda, MD, USA).

### 2.12. Statistical Analysis

All results are expressed as the mean + standard error of the mean (SEM). Data were processed using GraphPad software (http://www.graphpad.com) and analyzed for statistical significance using the Student’s *t*-test; a *p*-value < 0.05 was considered significant.

## 3. Results

### 3.1. Cx46 is Contained in EVs Released from MCF-7 Cells Overexpressing Cx46

Cx46 expression has been recently associated with tumor growth [13] and EVs, particularly exosomes, are considered key players in cancer development [25]. First, we evaluated the expression of Cx46 in MCF-7 cells and MCF-7 cells stably transfected with human Cx46 tagged with GFP in its C-terminus (MCF-7Cx46-GFP). Low levels of Cx46 were detected in MCF-7 cells (Figure 1A, left lane). In contrast, and as expected, MCF-7Cx46-GFP cells showed high levels of Cx46. Interestingly, two immunoreactive Cx46 bands were detected by Western blotting: a major band of apparent molecular mass (M_r_) of 70 kDa (which represents the sum of Cx46 and GFP molecular weights), and a minor band of M_r_ 50 kDa, probably representing Cx46 without GFP (Figure 1A, second column). In the case of MCF-7 cells, low levels of Cx46 were detected (Figure 1A, first column). Cx43 has also been involved in breast cancer progression [51], and also has detected in EVs isolated from Human Embryonic Kidney cell line (HEK) [52], a heart cell line (H9c2), and a retinal pigment epithelial cell line (ARPE-19) [20]. To evaluate the release of EVs in supernatants of MCF-7 and MCF-7Cx46-GFP cells and subsequently determined the presence of Cx46, we purified EVs by differential ultracentrifugation [53] from supernatants of cells cultured for 48 h in EV-depleted medium. First, we characterized our purified EVs by means of transmission electron microscopy. In both cell lines (MCF-7 and MCF-7Cx46-GFP) a heterogeneous populations of EVs was found, with a size and cup-shape expected to EVs; no apparent morphological differences between the EVs derived from the different cell lines were detected (Figure 2A). In both cell lines, the isolated EVs were between 100 to 200 nm in diameter, as evaluated by nanoparticle tracking analysis (NTA). The average EVs size was 140 nm, which is the expected size for EVs enriched in exosomes [54,55] (Figure 2B). In addition, overexpression of Cx46-GFP in MCF-7 cells increased the number of EVs released by 61.6 ± 2.0% compared to MCF-7 cells (Figure 2B,C).

To test whether Cx46-GFP was incorporated into the EVs, we characterized them by Western blotting. GM130 and calnexin, two cytoplasmic proteins, were detected in MCF-7 and MCF-7Cx46-GFP whole cell extracts (Figure 2D, left lanes), but they were absent in the EVs derived from these two cell lines (Figure 2D, right lanes). Thus, the isolated EVs were free of cytoplasmic protein contamination. In contrast, levels of Flotillin, Alix, and CD9, three protein markers of exosomes, were enriched in the isolated EVs from both cell lines, but they were not detected in the whole cell extracts (Figure 2D). Later, EVs were characterized for the presence of Cx46, and only showed Cx46 the EVs isolated from the MCF-7Cx46-GFP cell line showed Cx46 (Figure 2D). The Western blot pattern of immunoreactive bands was similar to that detected in whole cell lysates (Figure 1A, second lane). To further confirm the presence of Cx46 in EVs derived from MCF-7Cx46-GFP, an immunogold labeling was performed. We detected 10-nm gold particles which in close association with EVs membrane (Figure 2E, white arrows). Cx46 is a membrane protein that forms ion channels [56], therefore this suggests that Cx46 is embedded in EVs membrane.

### 3.2. The Presence of Cx46 on EVs Facilitates its Interaction/Internalization and Delivery of DNA in the Recipient Cell

Connexins are proteins responsible for cell-to-cell interaction [1], therefore we tested whether Cx46 contained in EVs enhanced the interaction between EVs and the surface of the recipient cells. To this end, we exposed MCF-7 and MCF-7Cx46-GFP cells to EVs with Cx46 (EVs-Cx46) or EVs without Cx46 (EVs-sansCx46). EVs were pre-stained with PKH26 to facilitate their visualization under confocal microscopy (Figure 3B,D). EVs interacted with the recipient cells independently of the presence of Cx46 in the recipient cell. At 37 °C, no significant difference in the number of spots (corresponding to EV-Cx46) per field between MCF-7 cells and MCF-7Cx46-GFP cells were detected. However, a 70 ± 20% decrease in the number of spots/fields was observed in both cell lines when incubated at 37 °C with EVs-sansCx46 (Figure 3A,C). These results imply that Cx46 contained in EVs improved the interaction between the EVs and the recipient cells.

To further characterize the participation of Cx46 in this process, we used lanthanum chloride (LaCl_3_) to block connexin hemichannels and any interaction between Cx46 and proteins on the recipient cell surface. The presence of La^3+^ caused a 70 ± 10% decrease in the number of EV-Cx46 interacting with MCF-7 or MCF-7-Cx46GFP cells (Figure 3B,D). These results suggest that Cx46 likely forming hemichannels is a relevant factor for the EV-recipient cell interaction. Taken together, these results strongly suggest that Cx46 hemichannels are important for the docking of EVs to the recipient cell plasma membrane in order to facilitate their endocytosis. In agreement with this idea, the incorporation of EVs inside the recipient cells was inhibited by 75 ± 9% when the incubation with MCF-7 and MCF-7Cx46-GFP cells was carried out at 4 °C (Figure 3B,D).

To confirm the role of Cx46 in facilitating the EV internalization, EVs-Cx46 and EVs-sansCx46 were loaded with a plasmid encoding GFP protein. These EVs were incubated with MCF-7 cells, which present low or non-detectable amounts of Cx46 expression (Figure 1A, left lane). Cells were incubated at 37 °C or 4 °C (to inhibit the internalization process), and, the GFP expression was measured by flow cytometry and fluorescence microscopy after 24 h. The results showed a more than 2-fold increase in the percentage of GFP positive cells of three times in MCF-7 cells incubated with EVs-Cx46 versus EVs-sansCx46 (9.5 ± 1.3% versus 3.6 ± 1.0%, respectively) (Figure 4A). As expected, the 4 °C incubation let to a decrease in GFP protein expression due to inhibition in internalization process (Figure 4A). To visualize the GFP expression in MCF-7 recipient cells, a representative image is shown (Figure 4B), first column represents the 37 °C incubation with EVs-Cx46 (botton) and EVs-sansCx46 (top), in the right column the 4 °C conditions are represented (Figure 4B).

To look in details the EVs-Cx46 docking and/or internalization, Z-stack confocal images were collected in MCF-7Cx46-GFP and MCF-7 cells (Appendix A).

### 3.3. EVs Containing Cx46 increase Migration/Invasion in MCF-7 Recipient Cells

Cell migration and invasion are key procedure involved in many biological processes including cancer. To evaluate whether EV-Cx46 affected the migration process of MCF-7 recipient cells, we exposed cells to EVs-Cx46 or EVs-sansCx46 and subjected them to a wound closure migration assay [57]. At 16 h an increase in closure area was observed in MCF-7 cells that were exposed to EV-Cx46 (41.3 ± 4.1%) compared to cells MCF-7 cells that were exposed to EVs-sansCx46 (32 ± 3.6%) (Figure 5A,B). No significant difference was observed regarding the untreated condition when MCF-7 cells were incubated with EVs-sansCx46 (29.6 ± 1.5% versus 32 ± 3.6%). The closing area of the cells that overexpress Cx46-GFP was greater than all the conditions tested (48 ± 6.5%) (Figure 5A,B). No difference in growth rate between the cell lines was observed (data not shown). Invasion is another important characteristic in cancer metastasis. To evaluate the possibility that EVs-Cx46 affected the invasiveness of the recipient cells, we performed a transwell cell invasion assay in MCF-7 cells exposed to EVs-Cx46 or EV-sansCx46. No significant differences were observed in the number of MCF-7 cells that crossed the Geltrex matrix in absence of any treatment (18 ± 3 cells/fields) compared to cells exposed to EV-sansCx46 (22 ± 2 cells/fields). However, when MCF-7 cells were incubated with EVs-Cx46, a significant increase in the number of cells that crossed the matrix was observed (62 ± 8 cells/fields) (Figure 5C,D); this number was even greater than that observed in MCF-7Cx46-GFP cells (45 ± 11 cells/fields). Together, these results show that EVs-Cx46 produces an increase in the migration and invasion of MCF-7 cells with a greater effect on cell invasion. Future research is necessary to determine the role of Cx46 on the cargo of EVs, and its possible mechanism in the progression and metastasis of breast cancer.

## 4. Discussion

In this study, we found that the expression of Cx46 in the MCF-7 breast cancer cell line increased the number of EVs released to the milieu and that these EVs contained Cx46. The presence of Cx46 in EVs enhanced the recipient cell internalization and interaction, facilitates the transfer of biological information between EVs and recipient cell, and migration of MCF-7 recipient cells. These results suggest that expression of Cx46 in cancer cells increases not only their metastatic properties but also the aggressiveness of neighboring cells, via release of EVs that contain Cx46.

Exosomes are normally released by diverse cell types including, fibroblasts, immune cells, neurons, tumors cells [31,33]. However, the frequency of release increases in certain pathological conditions, such a cancer [27,58].

To visualize and determine morphological differences between the EVs released by MCF-7 and MCF-7Cx46-GFP cells, transmission electron microscopy (TEM) was performed. Our results show that both cells release EVs with a typical exosomal cup-shaped morphology [59,60] (Figure 2A), and no morphological differences were observed between the EVs evaluated. The EVs size distribution observed with the NanoSight fall in the range of 80 to 200 nm, with a peak of 140 nm, no larger particles were observed, which would indicate that there is no formation of EV aggregates, and that the purification protocol used excludes contamination of large EVs. The reduction in EV size observed when comparing the results obtained by TEM and NTA has been described in samples prepared for TEM visualization [61]; it probably results from dehydration of the EVs after fixation for TEM analysis. No difference in the size distribution between EVs-Cx46 and EVs-sansCx46 were observed (Figure 2B), however the number of EVs released (as shown by NTA analysis) increased twice in the case of EVs released by cells that overexpress CX46-GFP (Figure 2B,C). As mentioned above, the frequency in exosome release increases in pathological conditions, such a cancer. These increases could be associated with the relationship described between Cx46 and hypoxia resistance [12]. Thus, cells exposed to hypoxic conditions could enhance their Cx46 expression [12], which in turn could enhance the release of EVs. Concurring with this hypothesis, an HIF-1α-dependent mechanism in EVs release under hypoxic conditions has been reported [62,63]. For example, renal tubular cells increase the number of EVs released under hypoxic conditions, and this release is HIF-1α dependent. In addition, the HIF-1α knockdown in renal cell reduces the EVs release and loses the protective effect against renal tubular injury [64]. The possible relationship between Cx46/HIF-1α and the effect of Cx46 in cancer cell aggressiveness guarantees further studies.

On the other hand, glioblastoma cancer stem cells (CSC) express higher levels of Cx46, and the maintenance of CSC phenotype has been attributed to the presence of Cx46, demonstrating its tumorigenic role [11]. Furthermore, cancer cells release EVs that can regulate different functions in receptor cells such as the immune response, angiogenesis, and metastasis [31]. To determine the presence of Cx46 in EVs release from MCF-7 and MCF-7Cx46-GFP breast cancer cells, Western blot analysis was performed. Previously, diverse proteins have been identified in exosome membrane, adhesion molecules, such as tetraspanins (CD9, CD63, CD81), integrins, and membrane receptors [65], among others. Our characterization of the purified EVs revealed that they were positive for the exosomal markers; flotillin, alix, and CD9, and lacked contamination with cytoplasmic proteins, such as GM130 and Calnexin (Figure 2D), suggesting that they are mainly exosomes. Interestingly, Cx46 was only detected in EVs isolated from MCF-7Cx46-GFP cells, with two enrichment bands similar to those observed in cell lysates (Figure 1A). These bands of 50 kDa and 68 kDa correspond to Cx46 [39] and Cx46 fused to GFP, respectively. These data suggests that the GFP is somehow cut off leaving a Cx46 with its “normal” molecular weight. Finally, to visualize the presence of Cx46 in the EVs membrane, a TEM immunogold assay was performed; 10-nm gold beads corresponding to Cx46 were observed in the membrane of EVs-Cx46 (Figure 2E), corroborating the observed in the immunoblotting results (Figure 2C).

The presence of Cx46 in exosomes has not been previously described; however, another type of connexin subtype, Cx43, has been implicated in the transferring of information from the inside of the exosome to the recipient cells, and this interaction is facilitated when the recipient cells express Cx43 [20,66]. In contrast, our results show that the presence of Cx46 in EVs increase the amounts of EVs detected in recipient cells, and this effect is independent of the presence of Cx46 in target cells (Figure 3A–D). In agreement with this hypothesis, EVs-Cx46 was incubated with lanthanum chloride (LaCl_3_), which has been used as a hemichannel inhibitor in uptake of DAPI in HeLa cells transfected with Cx46 [67]. The results showed a decrease in number of EVs interacting with the recipient cells, similar to what was observed in the 4 °C conditions (Figure 3A–D). This suggests the possibility that Cx46 contained in EVs plays a role in uptake of EVs via internalization in recipient cell. Consistent with this, incubation with EVs-sansCx46 produces a decrease in spots/fields similar to that obtained in the presence of lanthanum chloride (Figure 3A,C). Recently, a newer Cx46 inhibitor candidate has been described, Clofazimine; however no inhibitory effect on Cx46-hemichannels has been observed [46]. Therefore, Clofazimine was not used to inhibit the entry or fusion of EVs-Cx46 in recipient cells.

Thus, it is clear that the presence of Cxs, probably forming hemichannels allow EVs to somehow interact more efficiently with recipient cells and/or help to transfer and/or select which “information” is transferred from them to the cells. In this regard, EVs containing Cx43 present different types of miRNA compared to those without Cx43 [20,68]. To address the role of Cx46 contained in EVs in the transfer and incorporation of biological material in recipient cell, a GFP plasmid was incorporated into EVs-Cx46 and EVs-sansCx46. The percentage of MCF-7 GFP positive cells increased to more than double in presence of EVs-Cx46 compared to EVs-sansCx46 (Figure 4A,B), corroborating that the presence of Cx46 in the EVs facilitated somehow the fusion and entry and the consequent delivery of biological material from inside the EVs into the recipient cells. Inhibitors such a lanthanum chloride for Cx46 were not used because the size of the GFP plasmid is greater than the pore of the Cx46 hemichannel [2]. To visualize the EVs-Cx46 internalization, confocal Z-stack images were collected at 0.5 μm intervals in MCF-7Cx46-GFP and MCF-7 incubated with EVs-Cx46. The confocal microscopy images showed the red spot corresponding to EVs-Cx46 that were internalized by MCF-7Cx46-GFP cells (Figure 6A), unlike MCF-7 cells in which EVs-Cx46 were mainly seen on their surface (Figure 6B).It is likely that Cx46 overexpression will change the properties of the cell membrane as described for Cx46 and other connexins [69] facilitating their internalization. For a more detailed visualization of EVs-Cx46 internalization, Z-stack images were collected (Appendix A).

Cell invasion and the ability of the tumor cells to migrate to others organs is the focus of research for many years. The acquisition of the more invasive phenotype is correlated with a more aggressive phenotype.

In this context, our results suggest that EVs-Cx46 may transfer the ability to increase the migration and in a greater manner the invasion behavior in recipient cells (Figure 6A–D). This effect can be produced by information transferred from the EVs-Cx46, the presence of Cx46 in exosomes could have changed the miRNAs pattern in exosomes, similar to that described in Cx43 [68]. Furthermore, EVs secreted by tumor cells have been related to the transfer of more invasive and migratory phenotypes in recipient cells [70,71,72]. In these contexts, a representative scheme has been included. Within this context, a representative scheme has been included (Figure 7). Another possibility is the transference of Cx46 from EVs to recipient cells, in this context αVβ_3_ integrin contained in exosomes is transferred from EVs to recipient cells, promoting a migratory phenotype [36]. However, using Western blot analysis it was not possible to detect the presence of Cx46 in MCF-7 recipient cells after incubation with EVs-Cx46 (data not shown).

Finally, the discovery of Cx46 contained in EVs opens the possibility for the development of a new biomarker if its presence is determined in different fluids in which the EVs have been detected. Their determination and the establishment of their presence with a stage in the development of breast cancer would help the search for new therapies in the fight against this disease. Thus, it is very important to prove this idea in order to define the role of Cx46 as a pro-tumorigenic factor. As far as we know, all the experiments measuring or modifying Cx46 proteins levels are consistent in showing that this Cx is involved in an enhanced EV release, migratory and invasive behavior, and cancer stem cell properties. Our results show that the presence of Cx46 in EV facilitates its incorporation by recipient cells, but also increased their migration and invasiveness. These results strongly suggest a new function for Cx46 contained in cancer cell-derived EVs in its role of establishing malignant and metastatic cancer cells.

## 5. Conclusions

In conclusion, our results suggest for the first time that Cx46 is contained in EVs, and that this expression is associated with the biological role of EVs in breast cancer cells. Cx46 contained in EVs facilitates the entry and transfer of biological cargo to the receptor cells, in addition to increasing the migration and invasion of the recipient cells. Future work should be aimed to understanding the role of Cx46 in determining the content of EVs, and their participation in the transfer of biological information through Cx46 hemichannels. Advances in this area could contribute to the design of biomarkers and future Cx46 inhibitors, which could be closely associated with the progression of cancer and its metastasis, or to the design of Cx46 inhibitors. 

## Figures and Tables

**Figure 1 biomolecules-10-00676-f001:**
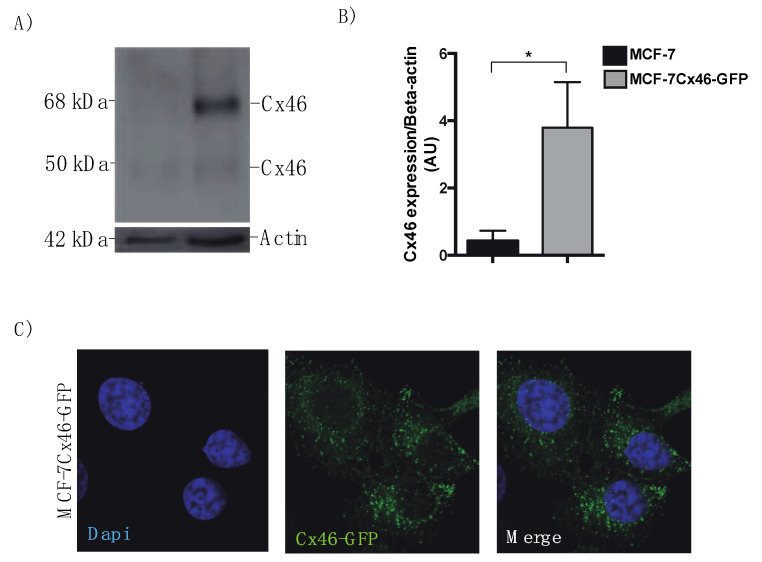
Connexin 46-GFP is overexpressed in MCF-7 cells. MCF-7 breast cancer cells were transfected with human connexin (Cx)-46 gene inserted in pcmV6-AC-GFP plasmid and selected by Neomicyn resistance. (**A**) Representative of three independent Western blots of MCF-7 and MCF-7Cx46-GFP cell lisates, immunoblots were incubated with anti Cx46, and anti Beta Actin (Santa Cruz Biotechnology; 1:2500). Protein bands were detected using Immobilon Forte western horse-radish protein (HRP) substrate and visualized with a LI-COR CDigit Systems. (**B**) The graph represents three densitometric independent analyses of the Cx46 and Beta Actin bands obtained using ImageJ software. Graphs were performed using GraphPad. Data are presented as the mean +/− SEM. * *p* < 0.05 (**C**) Representative fluorescence images of MCF-7Cx46-GFP cells Nuclei were visualized with Dapi (left), Cx46 was visualized with the GFP tag in the C-terminal portion of Cx46 (middle). Images were obtained using a Nikon Eclipse Ti-U inverted microscope.

**Figure 2 biomolecules-10-00676-f002:**
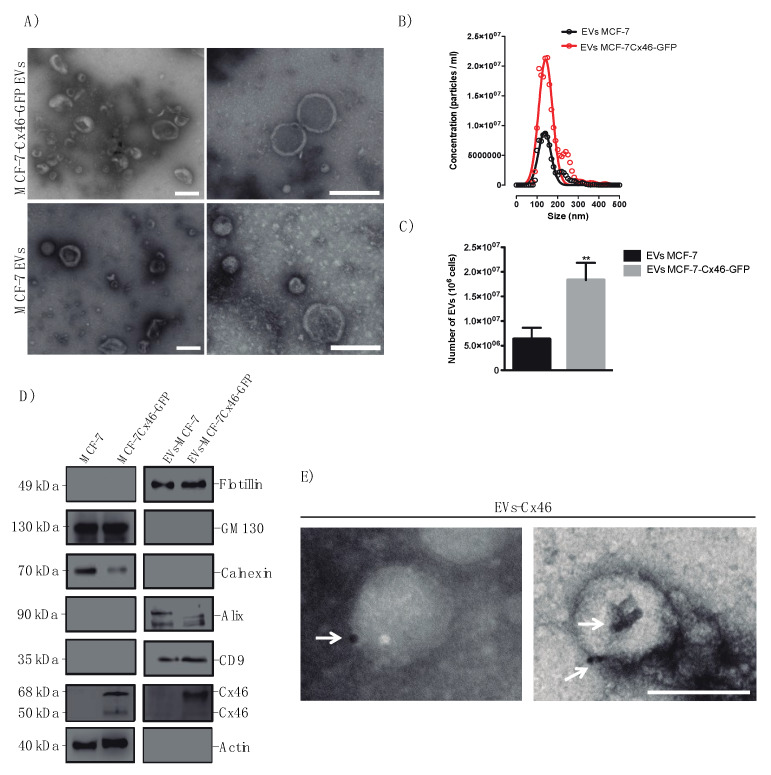
MCF-7Cx46-GFP derived EVs present Cx46 in their membrane. Purified exosomes by differential centrifugation were placed on Formvard carbon-coated grids, stained with uranyl acetate, and visualized under Transmission Electron Microscopy (TEM), in (**A**) Images show purified EVs released from MCF-7Cx46-GFP and MCF-7 cell. (**B**) Nanoparticle tracking analysis (NTA) of EVs secreted by MCF-7Cx46-GFP (red line) and MCF-7 (black line) cells; the mean size of particles detected was 140 nm. (**C**) Number of EVs released from MCF-7 and MCF-7Cx46-GFP cells, the number of EVs released was determined as the ratio NTA particles concentration per total number of cells at the time of purification. The results represent the average of three independent experiments ± SEM ** *p*-value < 0.05. (**D**) Immunoblot analysis of 30 μg of cell and EV lysates from MCF-7 and MCF-7-Cx46-GFP cell lines. The antibodies used for the immunoblot analyses are from top to bottom, Flotilin, GM130, Calnexin, Alix, CD9 (Cell Signaling), Cx46, and Actin (Santa Cruz Biotechnology). (**E**) Immunogold labeling of EVs released from MCF-7Cx46-GFP cells using anti-Cx46 antibodies, and a 10-nm gold particles-conjugated secondary antibodies. White arrows point to the gold particles. Scale bar 100 nm.

**Figure 3 biomolecules-10-00676-f003:**
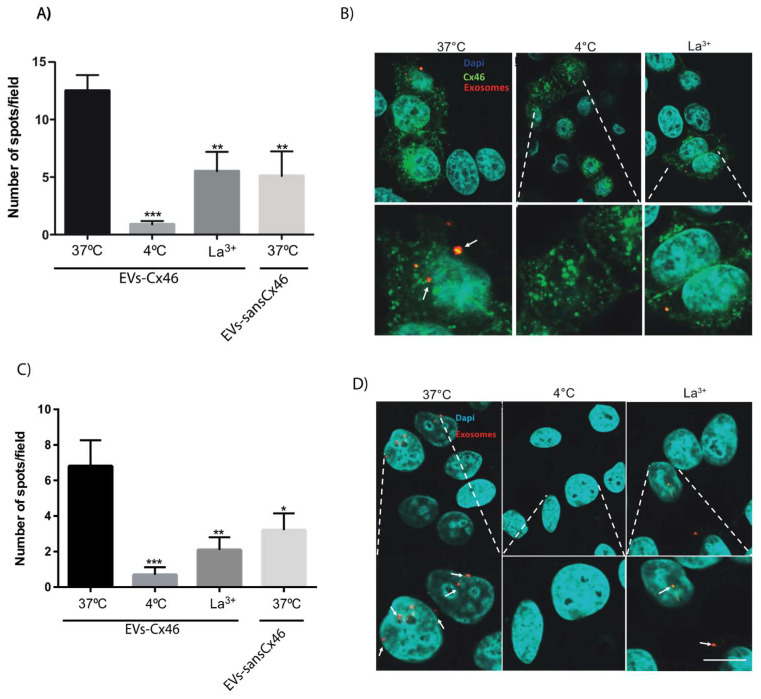
The presence of Cx46 in EVs facilitates the interaction with MCF-7Cx46-GFP and MCF-7 cells. Purified EVs-Cx46 or EVs-sansCx46 were stained with PKH26 (Sigma). Then, EVs-Cx46 were incubated with MCF-7Cx46-GFP (**A**,**B**) or MCF-7 (**C**,**D**) cells for 3 h in different conditions; 37 °C, 4 °C, 37 °C in the presence of lanthanum Chloride (LaCl_3_); EVs-sansCx46 were incubated with MCF-7Cx46-GFP (**A**,**B**) or MCF-7 (**C**,**D**) cells at 37 °C for 3 h. Cells were rinsed with PBS to eliminate unbound EVs, fixed with 4% of paraformaldehyde (PFA), mounted with DABCO (Fisherscientific), and visualized under confocal microscopy. (**A**,**C**) Graphs shows the EVs spots/fields average from ten different fields of the four conditions evaluated in MCF-7Cx46-GFP (**A**) and MCF-7 (**C**) cells. (B, D) Representative images illustrating the cells under the first three conditions showing, Cx46-GFP in green, EVs-Cx46 in red (white arrow), and nuclei in blue. Data are presented as mean ± SEM (n = 3); * denotes *p* < 0.05, ** *p* < 0.01 and *** *p* < 0.001. Scale bar, 25 µm.

**Figure 4 biomolecules-10-00676-f004:**
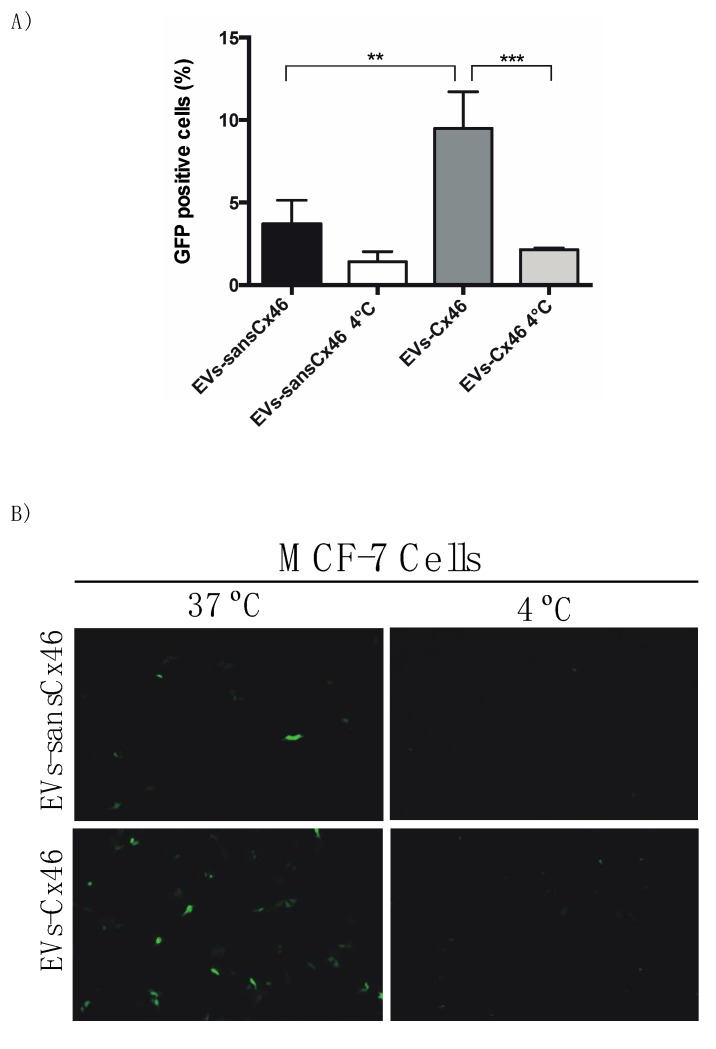
Cx46 contained in EVs facilitates the DNA delivery in the recipient cell. A GFP plasmid was incorporated in EV-Cx46 and EVs-sansCx46, and co-cultured with MCF-7 recipient cell for 45 min. Then, the, medium was replacement with EVs-depleted medium. After 24 h and the expression of GFP protein was evaluated by flow cytometry and visualized under fluorescence microscopy. (**A**) Graph represents the percentage of GFP-positive cells measure by flow cytometry. Data are presented as the mean of three independent experiments ± SEM; **, *p* < 0.01 and ***, *p* < 0.001 (**B**) Representative images of GFP expression in MCF-7 cells at 37 °C and 4 °C conditions.

**Figure 5 biomolecules-10-00676-f005:**
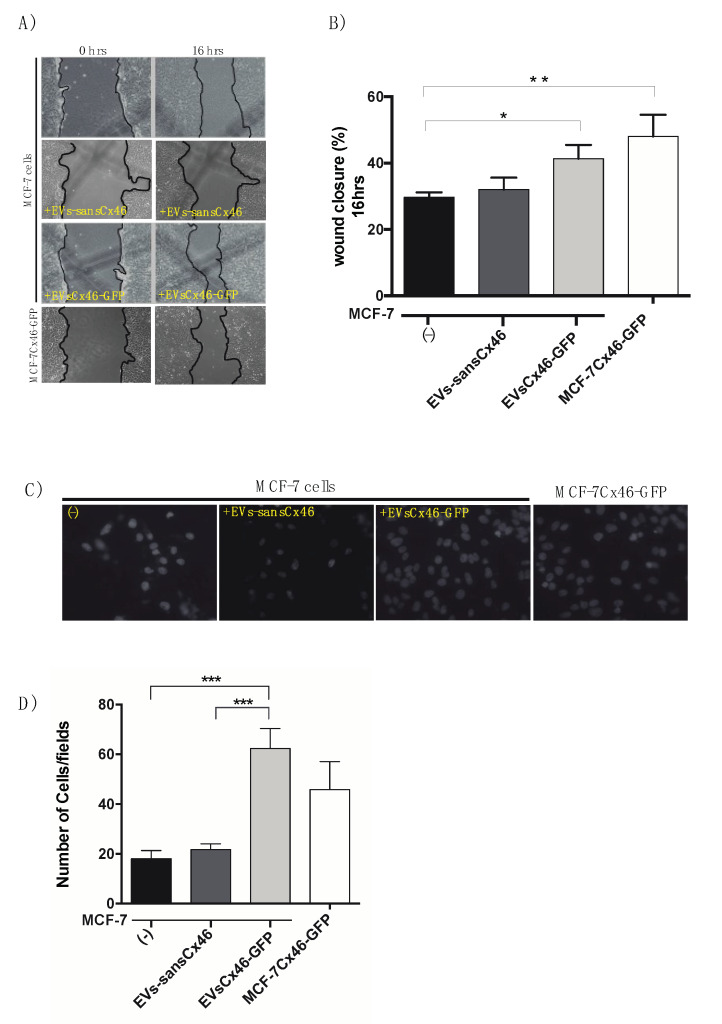
Cx46 contained in EVs increase migration/invasion in MCF-7 recipient cells. MCF-7 cells were co-cultured with EVs-Cx46 and EVs-sansCx46. (**A**) Representative images of time course of cell migration wound healing invasion assays at 0 and 16 h, using MCF-7 as the recipient cell line with two donors EVs (EVs-Cx46 and EVs-sansCx46), yellow highlighter letter indicate the kind of EVs donor. (**B**) Graphical representation of closing area percentages of wound healing of three independent experiments at 16 h. Errors were calculated from wound closure at each time point and normalized to the wound closure area at the initial time point (0 h). (**C**) MCF-7 matrigel invasion assay, cells were seeded in serum free media on a matrigel-coated filter and their migration was evaluated in presence of EVs-Cx46 and EVs-sansCx46, MCF-7 without EVs, and MCF-7Cx46GFP was used as a control. Migration was quantified by microscopic visualization. (**D**) Graphic represents the average total number of invasive cells from 10 different fields. *p*-values were calculated using two-sided Student’s *t*-tests. (n = 3). Data are means ± SEM (n = 3); * denotes *p* < 0.05, ** *p* < 0.01 and *** *p* < 0.001.

**Figure 6 biomolecules-10-00676-f006:**
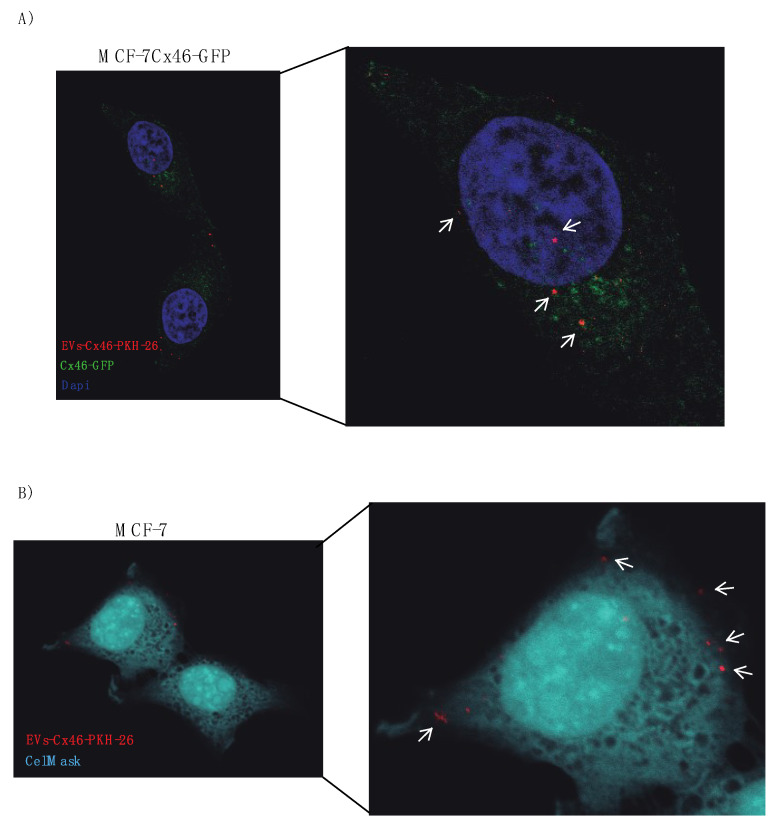
Cx46 contained in EVs facilitate the internalization in recipient cell. Similar amounts of EVs-Cx46 and EVs-sansCx46 were labeled with PKH26 and added to MCF-7Cx46GFP or MCF-7 cells. Representative images of EVs-Cx46 interacting with MCF-7Cx46-GFP (**A**) and MCF-7 (**B**) cells, visualized under confocal microscopy. Arrows indicate the EVs-Cx46 PKH26-stained. Plasma membrane stains CellMask was used to visualize MCF-7 cell. The magnification is shown in the right panel. Scale bar 25 µm.

**Figure 7 biomolecules-10-00676-f007:**
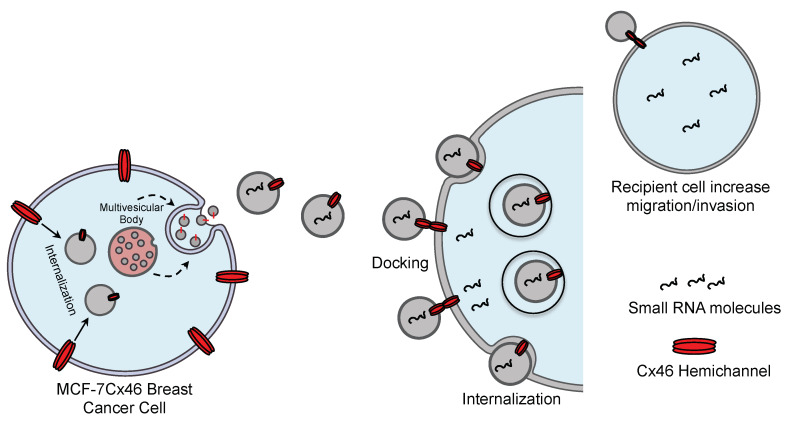
Squematic role of Cx46 contained in EVs mediating intercellular communication. The intraluminal vesicles from MCF-7Cx46 cells, mature into multivesicular bodies, these can fuse with the plasma membrane and release EVs with Cx46 into the extracellular space. Cx46 contained in EVs can interact with the surface of recipient cell surface and then being internalized by the recipient cell. The presence of Cx46 in the membrane of EVs may facilitate their docking and internalization and therefore accelerate the release of EVs-Cx46 content (small RNA molecules) into the recipient cell causing an increase in migration and invasion in the recipient cell. This process is facilitated by the presence of Cx46 in the recipient cell, but is not entirely dependent.

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
