# Peer review of "Connexin-46 Contained in Extracellular Vesicles Enhance Malignancy Features in Breast Cancer Cells"

_biomolecules, 2020, doi:10.3390/biom10050676_

Round 1

Reviewer 1 Report

In this manuscript, Acuña et al. investigate on a Connexin-46 Contained in Extracellular Vesicles, which relate to the tumor growth in Breast Cancer Cells. The authors also nicely compared with Xenograft mice model. The article is very interesting, and the work is very appropriate in scope and length for publication in the Biomolecules journal. However, before its acceptance, the following question should be fully addressed. Please provide a schematic diagram in detail in the main manuscript. I was unable to open the supporting information using the mentioned link https://www.mdpi.com/xxx/s1 Please send the supporting information separately

Author Response

Response to Reviewer 1 Comments

Point 1: Please provide a schematic diagram in detail in the main manuscript.

Response: The schematic model is detailed, and incorporated in the main manuscripts the proposed model is attached as a pdf file.

Point 2: I was unable to open the supporting information using the mentioned link

Response: Done

Reviewer 2 Report

In this revised version of the manuscript titled “Connexin-46 Contained in Extracellular Vesicles Enhance Malignancy in Breast Cancer Cells” by Rodrigo A. Acuña et al., the authors addressed/responded to all the issues raised in the previous stage. Moreover, the presentation of the work gained the clarity and intelligibility required for a full assessment of the paper, and more importantly, it reached the suitability for publication in “Biomolecules” journal.

Author Response

Thanks a lot for your comments 

Reviewer 3 Report

The Authors of the publication responded satisfactorily to my comments. The publication in its current form deserves to be published in Biomolecules

Author Response

Thanks a lot for your comments 

This manuscript is a resubmission of an earlier submission. The following is a list of the peer review reports and author responses from that submission.

Round 1

Reviewer 1 Report

The authors generate MCF-7 cells that overexpress Cx46 and  isolate EVs from these cells to show that Cx46 is present in EVs. The authors show that C46 containing EVs will promote cell migration and conclude that Cx46 release is inherent to this process. This finding is unique and interesting but the study lacks mechanistic insight. The following improvements are suggested:

Major:

-While the isolated EVs are shown to have Cx46, it is not clear whether Cx46 is present in the membrane to transmit the contents of the EV or as a direct factor for promoting cell migration.

-Is this only inherent to MCF-7 cells or ER+ breast cancers or is this consistent with other types of breast cancer subtypes? Along the same lines, what is the expression of Cx46 across breast cancer subtypes? Has anyone looked? In the least, these points should be discussed.

-Where is the GFP tag on Cx46 (N-terminus or C-terminus) and might this effect the results? The study does not even use the GFP tag to visualize Cx46 in any way. A non-tagged version should really be used since prior studies have shown that placement of the GFP tag can greatly alter Cx function.

Minor:

-At line 60 the word "xerograph" should be replaced to "xenograft"

-The size of the figure labels in 3A need to be increased.

Reviewer 2 Report

In this manuscript entitled "Connexin-46 Contained in Extracellular Vesicles Enhance Malignance Characteristics in Breast Cancer Cells", by  Rodrigo A. Acuña et al., the authors hypothesised the presence of Connexin-46 (Cx46) in extracellular vesicles (EVs) isolated from the MCF-7 breast cancer cell line. They measured the amount of EVs both in wild-type MCF-7 cell line and in MCF-7 cells with overexpression of Cx46 linked to GFP (MCF-7-Cx46-GFP). They reported that overexpression of Cx46-GFP increases the number of EVs released in the supernatant of the cells compared to the MCF-7 cells cultured under control condition. Western-blot analysis was then used by authors to confirm the presence of Cx46 in EVs. By evaluating EVs-cell interaction, they found that EVs incorporation in recipient cells was enhanced by the presence of Cx46 at the surface of EVs. Finally, by wound healing and transwell assays, they reported that Evs-Cx46 increase migration and invasion of MCF-7 breast cancer cell line, but not the expression of Cx46.

Overall, the experimental design is well conceived by the authors, and the findings on Cx46 are characterised by flashes of novelty. However, a pair of critical issues still exist and deserve to be addressed by the authors.

- Since it is known that Cx46 expression is linked to the hypoxia conditions and that hypoxia is a key inductor of EVs release from cancer cells, the reader expects to see experiments about the presence of Cx46 in EVs released from MCF-7 cell cultured under hypoxic conditions, and not only in MCF-7 cells which overexpress Cx46 linked to GFP

- The discussion section is arranged in a way that it seems like a “wordy” results replica. It must be improved, and it should relate much more the study findings with those of other similar studies

Minor issues

- Figure 1A: this plot does not seem to be an immunogold figure. In the bottom right panel (i.e., MCF-7-EVs) is shown a white arrow that is supposed to show gold particles attached to Cx46. It appears like the white arrow was misplaced here

- Figure 2B and Figure 2D: the name of the cell lines (i.e., MCF-7, MCF-7-Cx46-GFP, or HeLa) used by authors to isolate the EVs should be reported on these plots

- Some typo errors are present throughout the manuscript and need to be fixed by the authors

Reviewer 3 Report

The study has an interesting methodological aspect.
The authors write a too vague conclusion in the summary. The parameters tested can not be used as a clinical marker, due to the lack of objective and non-standardized measurements.
In the part of the article 'Material and methods', the HeLa line is the research material. The authors do not present research results on these cells.
There are also no explanations of abbreviations, eg MCF-7-Cx46-GFP.

Reviewer 4 Report

The manuscript by Acuña et al shows the effect of Cx46-EVs in promotion of cell migration and invasion in a breast cancer cell line, but lacks some studies to achieve stronger conclusions.

- Authors should provide some database analysis of the correlation of high Cx46 levels and breast cancer prognosis (check cancer databases), as it is not clear why Cx46 could be used as a new biomarker in breast cancer progression.

- They make reference to Banerjee et al. (2010) research where they found Cx46 overexpression in MCF7 cells. It would also be very interesting to check Cx46 in EVs derived from hypoxic MCF7, and Cx46-EV effect on hypoxia resistance in MCF7.

- Cx46-overexpressing MCF7 cells and Cx46-EV need to be characterized in terms of proliferation and colony formation.

- MCF7 cells need to be treated with Cx43-EVs to see if they  these vesicles further increase migration, invasion, proliferation or colony formation. The authors could treat a fibroblast cell line to confirm the effect of Cx46-EVs on promoting migration and invasion.

- It is not clear why the authors use EVs derived from Hela cells as a Cx46-negative control instead of just using EVs derived from the wild type MCF-7 cells, which (as showed in Fig. 1D) do not express Cx46 and do not express Cx43.

Minor revisions:

- In Fig. 2A and 2C it is very difficult to appreciate the PKH26 positivity. Authors should used better representative images or increase the magnification.

- How is prepared the depleted medium used for EVs isolation? It should be mentioned in the test (line 84) or in the methodology section.

- Authors used lanthanum as a hemichannel blocker (line 127), but later they described it as a general gap junction inhibitor (line 216). Clarified this.

- Authors should provide a clear explanation of why the effect of Cx46-EVs is greater than Cx46 overexpression by itself.

- Abbreviations: GJ better than GJC and GJIC better than intercellular GJC communication

- Pg. 1, sentence 44 onwards: add comas to facilitate the reading

Pg. 2, sentence 49: instead of/besides reference 11, I would add the exact reference of the Cx43 in exosomes article

Pg. 3, sentence 98: enriched

Fig. 1A: what is the white arrow?

Pg. 7, sentence 190: cancer [ref?]

Pg. 7, sentence 209: “together the results confirm for the first time the presence of Cx46 in EVs isolated from breast cancer cells” -> It should be reformulated, as these are modified (Cx46-overexpressing) cancer cells, not the original ones

There are references lacking in the discussion, for ex. Pg. 7 sentence 212

In line 141 it is written “then” when it should be “the”.

In line 214, it is written “independent at the presence”, when it should say “independent of the presence”.

Summary: revise the sentences, some of them lack subject. Also throughout the whole text (example: “On the other hand is known…”).

Pg. 7, sentence 216: gap junction or hemichannel inhibitor?- Authors should provide some database analysis of the correlation of high Cx46 levels and breast cancer prognosis (check cancer databases), as it is not clear why Cx46 could be used as a new biomarker in breast cancer progression.

- They make reference to Banerjee et al. (2010) research where they found Cx46 overexpression in MCF7 cells. It would also be very interesting to check Cx46 in EVs derived from hypoxic MCF7, and Cx46-EV effect on hypoxia resistance in MCF7.

- Cx46-overexpressing MCF7 cells and Cx46-EV need to be characterized in terms of proliferation and colony formation.

- MCF7 cells need to be treated with Cx43-EVs to see if they  these vesicles further increase migration, invasion, proliferation or colony formation. The authors could treat a fibroblast cell line to confirm the effect of Cx46-EVs on promoting migration and invasion.

- It is not clear why the authors use EVs derived from Hela cells as a Cx46-negative control instead of just using EVs derived from the wild type MCF-7 cells, which (as showed in Fig. 1D) do not express Cx46 and do not express Cx43.

Minor revisions:

- In Fig. 2A and 2C it is very difficult to appreciate the PKH26 positivity. Authors should used better representative images or increase the magnification.

- How is prepared the depleted medium used for EVs isolation? It should be mentioned in the test (line 84) or in the methodology section.

- Authors used lanthanum as a hemichannel blocker (line 127), but later they described it as a general gap junction inhibitor (line 216). Clarified this.

- Authors should provide a clear explanation of why the effect of Cx46-EVs is greater than Cx46 overexpression by itself.

- Abbreviations: GJ better than GJC and GJIC better than intercellular GJC communication

- Pg. 1, sentence 44 onwards: add comas to facilitate the reading

Pg. 2, sentence 49: instead of/besides reference 11, I would add the exact reference of the Cx43 in exosomes article

Pg. 3, sentence 98: enriched

Fig. 1A: what is the white arrow?

Pg. 7, sentence 190: cancer [ref?]

Pg. 7, sentence 209: “together the results confirm for the first time the presence of Cx46 in EVs isolated from breast cancer cells” -> It should be reformulated, as these are modified (Cx46-overexpressing) cancer cells, not the original ones

There are references lacking in the discussion, for ex. Pg. 7 sentence 212

In line 141 it is written “then” when it should be “the”.

In line 214, it is written “independent at the presence”, when it should say “independent of the presence”.

Summary: revise the sentences, some of them lack subject. Also throughout the whole text (example: “On the other hand is known…”).

Pg. 7, sentence 216: gap junction or hemichannel inhibitor?

- Discussion should be rechecked for grammatical and ortographic errors. It should be discussed what kind of molecules might be being recruited by Cx46-EVs that promote increased invasion and migration.